# Occupations on the map: Using a super learner algorithm to downscale labor statistics

**Michiel van Dijk**[1,2]*, **Thijs de Lange**[1], **Paul van Leeuwen**[1], **Philippe Debie**[3]

**1** Wageningen Economic Research, the Hague, the Netherlands, **2** International Institute for Applied Systems Analysis, Laxenburg, Austria, **3** Marketing and Consumer Behaviour Group, Wageningen University, Wageningen, The Netherlands

* michiel.vandijk@wur.nl

**Data Availability Statement:** The main results and input datasets of this study are presented in the Supporting information files. All input and output maps produced for this study can also be accessed by means of an interactive web application (https://

## Abstract

Detailed and accurate labor statistics are fundamental to support social policies that aim to improve the match between labor supply and demand, and support the creation of jobs. Despite overwhelming evidence that labor activities are distributed unevenly across space, detailed statistics on the geographical distribution of labor and work are not readily available. To fill this gap, we demonstrated an approach to create fine-scale gridded occupation maps by means of downscaling district-level labor statistics, informed by remote sensing and other spatial information. We applied a super-learner algorithm that combined the results of different machine learning models to predict the shares of six major occupation categories and the labor force participation rate at a resolution of 30 arc seconds (~1x1 km) in Vietnam. The results were subsequently combined with gridded information on the working-age population to produce maps of the number of workers per occupation. The super learners outperformed (n = 6) or had similar (n = 1) accuracy in comparison to best-performing single machine learning algorithms. A comparison with an independent high-resolution wealth index showed that the shares of the four low-skilled occupation categories (91% of the labor force), were able to explain between 28% and 43% of the spatial variation in wealth in Vietnam, pointing at a strong spatial relationship between work, income and wealth. The proposed approach can also be applied to produce maps of other (labor) statistics, which are only available at aggregated levels.

## Introduction

Labor is recognized as one of the three primary factors of production in economics and accounts for around 50% of total global income [1]. Work forms a central part of most people's lives. Globally more than 3.1 billion people are working actively or are looking for work [2] and those that are employed spend around 40 hours per week on the job [3]. There is overwhelming evidence that labor activities are distributed unevenly across space because of a combination of differences in costs, economies of scale and spillover effects [4–8]. However, detailed statistics that illustrate the geographical distribution of labor and work are not readily available. ILOSTAT (ilostat.ilo.org), the most comprehensive global labor statistics database

shiny.wur.nl/occupation-map-vnm). Input and output data are available on Zenodo (DOI: 10.5281/zenodo.6419272) and scripts to reproduce the analysis are available on GitHub (https://github.com/michielvandijk/occupations_on_the_map).

**Funding:** This research was funded by a grant Wageningen University & Research Programme on "Food Security and Valuing Water" (project code KB-35-005-001) that is supported by the Dutch Ministry of Agriculture, Nature and Food Quality, and a contribution from the Wageningen University and Research investment fund. The funders had no role in study design, data collection and analysis, decision to publish or preparation of the manuscript.

**Competing interests:** The authors have declared that no competing interests exist.

maintained by the International Labor Office (ILO), only provides country-level data. Summary reports of population censuses and national labor-force surveys sometimes present subnational labor information, but mostly at the level of coarse first-level administrative units. The availability of spatially explicit labor statistics will support the formulation of targeted social policies that aim to improve the match between local labor supply and demand, and support local employment, contributing to economic growth and welfare.

The contribution of this paper is to demonstrate an approach to downscale subnational labor statistics to a fine-scale spatial grid using machine learning approaches. As such, this paper contributes to a rapidly expanding literature, which has used machine learning and advanced statistical models to create fine-scale gridded maps of socio-economic indicators. Key examples include the mapping of population [9], educational attainment [10], child growth [11], poverty [12] and wealth [13]. To the best of our knowledge, this is the first application to apply these techniques to downscale labor information.

Our approach resembles that of [14] and [15], who created gridded population and livestock maps, respectively. In contrast to these papers, which applied a single machine learning model (i.e. random forest), we used an ensemble approach, a so-called super learner, in combination with high-resolution remote sensing data and other spatial predictors to predict the shares of six major occupation categories and the labor force participation rate at a resolution of 30 arc seconds (~1x1 km) in Vietnam. A super learner combines the results of different machine learning models to generate predictions with the same or higher accuracy than those of single models [16]. Apart from better performance, combining the outcomes of multiple machine learning models also results in more robust outcomes [17], which is particularly important in case machine learning techniques are used to extrapolate results [18], such as in our application.

The results of our analysis provide insights into the geographical distribution of workers within a country. The occupation maps make it possible to identify regions that are characterized by low-skilled employment, which might be vulnerable to rising imports from low-cost countries [19]. Similarly, they can be combined with climate change projections to show the locations where workers will be most exposed to extreme temperatures, which is particularly relevant for active workers in the agriculture, construction, and manufacturing sectors [20]. This type of information can be used to better assess the impact of climate change on labor productivity and associated losses in national income [21, 22]. The occupation maps can also be regarded as proxies for the geographical distribution of income in a country as there is a strong correlation between occupational attainment and wages [23]. Finally, the maps also provide broad information on the spatial distribution of industrial activity as workers tend to live close to the workplace [24, 25], and the occupation categories are closely related to the main economic sectors: agriculture, manufacturing and services [26].

We selected Vietnam as a case study to demonstrate our approach because it is a lower middle-income country that has experienced a phase of rapid economic growth and structural transformation [27]. As a consequence, the occupational distribution is rather diverse, including roughly equal shares of agricultural and non-agricultural workers as well as a fair share of high-skilled jobs (S1 Fig). Another advantage was the availability of relatively detailed subnational occupation statistics, which were an essential input for the analysis.

## Materials and methods

Fig 1 summarizes our analytical approach. The next sections provide additional information on the target variables and their data sources, the super learner algorithm that was used to downscale the subnational labor statistics and the geospatial predictors that informed the model.

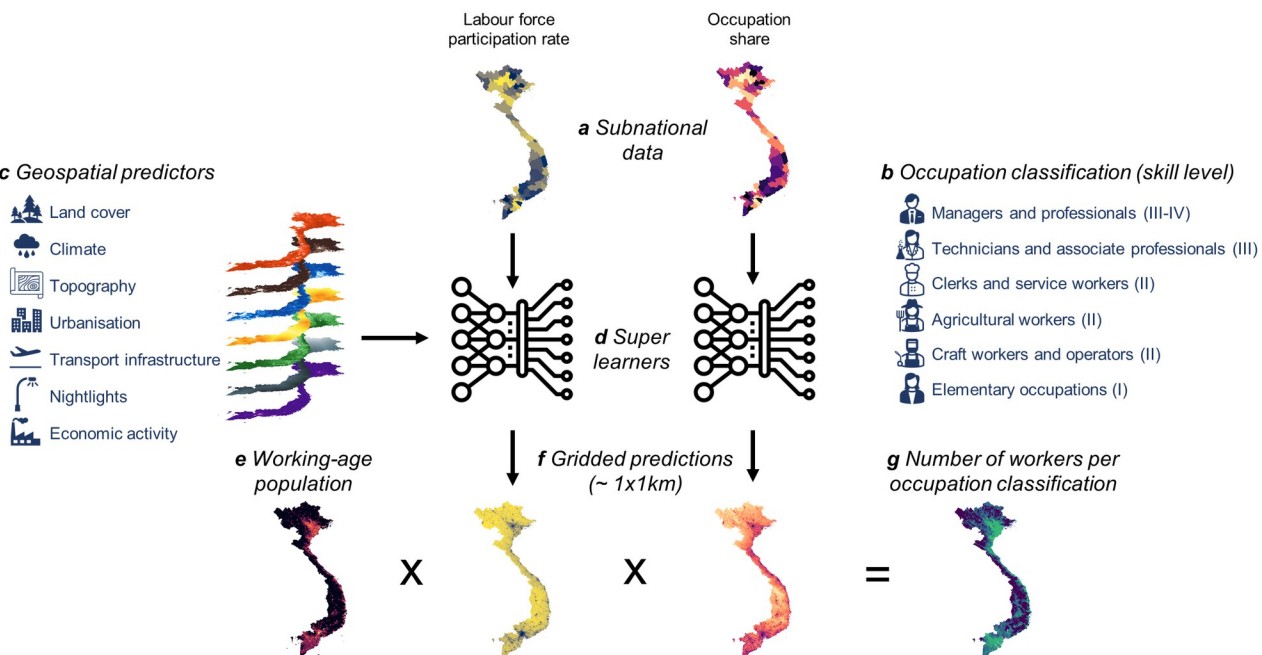

**Fig 1. Overview of the analytical approach.** (a) Subnational statistics on the labor force participation rate and occupation shares for (b) six occupation classes and (c) a large number of geospatial predictors were combined to train (d) super learner models. (e) Spatial information on the working-age population was combined with (f) resulting gridded predictions to produce (g) maps of the number of workers per occupation category. This figure has been designed using resources from Flaticon.com. Maps produced using data from [30] and calculations by authors, see text.

## Target variables

Our primary target variable was the total number of workers within an occupation category. We distinguished between six broad occupation categories based on the most recent version of the International Standard Classification of Occupations (ISCO) (S1 Table). The ISCO occupation classes are designed to capture two interrelated concepts: (a) the type of job since occupation is defined as a "set of jobs whose main task and duties are characterized by a high degree of similarity" and (b) skill level, which is defined as "function of complexity and range of tasks and duties to be performed in an occupation" [28]. To estimate the number of workers $O_i$ per occupation category $i = 1,\ldots, 6$, we broke it down into three components, in line with standard ILO definitions (ilostat.ilo.org):

$$O_i = WP \times LFPR \times OS_i$$

where *WP* is the working-age population, which, is defined as all persons aged 15 and older. *LFPR* is the labor force participation rate, defined as the number of persons in the labor force as a percentage of the working-age population. The labor force is the sum of the number of persons employed and unemployed, aged 15 and older. *OS* is the share of the labor force with occupation *i*.

The three occupation components have a distinct spatial distribution (S2 Fig). Labor force participation rates are higher in remote areas and lower in the major cities, such as Hanoi and Ho Chi Minh City, and the more densely populated coastal areas. This is consistent with the patterns in other countries, which show that labor force participation, in particular for women, is higher in rural areas because of limited possibilities to participate in educational activities and more possibilities to combine child-rearing and farm work in the absence of off-

farm employment [29]. Similarly, the share of non-agricultural occupations and the size of the working-age population tend to be higher in densely populated urban areas.

Our main source of information for the labor statistics was the 2009 Population and Housing Census organized by the Vietnam General Statistics Office that is available from IPUMS [30]. Representative labor statistics were available for 674 districts, which are the second-level administrative units in Vietnam. The census included questions for each person in the labor force on his/her type of occupation using the detailed ISCO 08 classification. We used this information to calculate the share of each major occupation group at the district level. Similarly, we also used the census to estimate the district labor force participation rate by dividing all persons that are categorized as being in the labor force by the total population aged 15 and older.

To calculate the working age population at each grid cell, we used population maps with 5-year age group compositions from [31]. The main data source for these products was the 2009 Vietnam population census, which we also used as the main input.

As a final step, we used a logit transformation to change the scale of our dependent variables, which are both proportions that are bounded between zero and one. As the model predictions are not guaranteed to be within this range, it is recommended to change the scale to between negative and positive infinity [32].

## Geospatial predictors

Initially, we selected 32 predictors, which we expected to explain the observed spatial distribution of the labor force participation rate and occupation categories (S2 Table). The majority of predictors were taken from the WorldPop (worldpop.org) and WorldClim (worldclim.org) open access archives, which contain a diverse range of remote sensing and other geospatial data at a resolution of 3–30 arc seconds that cover multiple periods, including land cover, night lights, transport networks, topography and climate indicators. In addition, we added geospatial layers with information on distance to large industrial facilities (energy, iron, steel and cement plants, and mines) and major transport hubs (airports and ports), which are often part of industrial zones, where a large number of people are employed. As workers tend to live in the vicinity of their work, we assumed that these layers have predictive power and support the training of the machine learning models. Information on the location of power, iron and steel, and cement plants was taken from the Global Infrastructure Emissions (GID) database (www.gidmodel.org.cn). The GID contains information on emissions from energy-intensive industries, which can be regarded as a proxy for their location. We also collected information on the location of mining areas, airports and ports. These datasets were further processed to create geospatial layers with distance information.

Where possible, we selected layers with information for the year 2009, consistent with the population census data. However, in a few cases, in particular for the predictors related to the location of industrial facilities, only recent data was available. To harmonize the data, all raster layers were resampled to a resolution of 30 arc seconds in WGS 84 projection. As the occupation data was available at the second administrative unit level, all predictors were aggregated to the same level using the median values for each administrative area.

We applied three feature engineering steps that result in better performance of machine learning models [32]: (a) we applied a Yeo-Johnson power transformation [33] to make the variable distributions more symmetric, (b) we normalized all data to have a standard deviation of one and a mean of zero, where mean values and standard deviations were estimated from the training dataset and (c) we removed all variables that had a correlation of 0.7 or larger with other variables. As a consequence of the last step 14 predictors were removed from the analysis (S3 Fig).

## Super learner

We used an ensemble approach, referred to as a super learner, to predict labor statistics at the grid level. A super learner is an algorithm that combines the results of multiple machine learning models. Predictions are then generated by weighting the outcomes of the individual member models. It has been demonstrated that predictions of a super learner have the same or higher accuracy in comparison to those generated by means of single machine learning models [16].

The process to train the super learner followed the steps as described in [16]. In total, we trained seven separate super learners; one model to spatially predict the labor force participation rate (*LFPR*) and six models to predict the shares for each occupation class (*OS_i*). Each super learner combined the results of six different machine learning algorithms that have been used extensively in (spatial) prediction exercises: random forest, (random_forest) extreme gradient boosting (xgboost), neural network (neural_network), polynomial (svm_poly) and radial (svm_radial) basis support vector machines, and generalized linear model via penalized maximum likelihood (glmnet). All six machine learning algorithms and the super learner models used data at the subnational level for training and testing (i.e. 674 data points in total).

All models (i.e. super learner and member models) were implemented using the tidymodels framework [34] in the R software environment [35]. We started by applying the tidymodels *tune* and *dials* packages to optimize the tuning of hyperparameters for all super learner member models. For each model, we conducted a grid search with 30 model-specific parameter combinations that were selected using a Latin hypercube design. Each parameter combination was evaluated and fitted by means of 5 repeats of 10 fold cross-validation, using 80% of the dataset (n = 538) for training. Inspection revealed that a few models had very poor performance, predicting (near) constant values. Therefore, we removed all models for which the predictions had a standard deviation lower than 0.001. We preferred to exclude these poor models as it results in more parsimonious super learner models, without loss in performance. In the next step, we tuned a regularized generalized linear model to optimally combine the predictions of the member models and determine the relative weights for the super learner. We used the default settings that constrain the coefficients of the super learner to be non-negative. We analyzed the accuracy of our modeling framework by fitting all selected member models (i.e. those with a weight larger than zero) and the super learner on hold-out data that contains 20% of the main dataset (n = 136). For each model, we derived the RMSE and the $R^2$. The super learners were combined with the 30 arc seconds predictor maps to predict the labor force participation rates and occupation shares for all grid cells in Vietnam. We applied an inverse logit transformation to obtain values between zero and one. For the predictions of the six occupation shares we used a standardization function to ensure that the total sum of shares equals one in each grid cell:

$$occupation_{ij}^c = \frac{occupation_{ij}}{\sum_{i=1}^{6} occupation_{ij}}$$

where $occupation_{ij}^c$ is the corrected share for occupation *i* in grid cell *j*. Finally, the occupation maps were produced by multiplying grid-level information on labor force participation rate, occupation share and working-age population. We followed the approach used by [18] to calculate 67% (1 standard deviation) grid-cell prediction errors for the super learner models. This involved two steps (a) fitting a quantile regression random forest model using the training dataset predictions of the super learner member as input and (b) estimating error statistics per grid cell using the *forestError* package that implements the method proposed by [36].

For each member model we created variable importance measures that demonstrated by how much a models' performance (measured by the RMSE) changes if an explanatory variable is removed, as well as accumulated local effects plots, which showed how features relate to the machine learning predictions on average. The methods to create both types of plots, implemented with the R *DALEX* package [37], are model-agnostic and did not assume anything about the structure of the super learner member models [38]. All maps were produced using the R software environment [35]. Scripts to reproduce the analysis are available at: https://github.com/michielvandijk/occupations_on_the_map.

## Results

### Number of workers per occupation

Fig 2 presents maps of the number of workers for each of the six occupation categories for the entire country, and around Hanoi and Ho Chi Minh City, the two regions with the highest (working age) population density (S2h Fig). The maps clearly show different spatial patterns between the occupations. A relatively large number of agricultural workers are located in the Red River (South-East of Hanoi) and Mekong (South-West of Ho Chi Minh City) River Deltas as well as in the rural areas throughout the country. The high absolute number of agricultural workers observed very close to Hanoi can be explained by the combination of a large working-age population, widespread rice cultivation and well-developed urban agriculture [39] in this area. As expected, the share of agricultural workers is very small (0–2%) in urban areas and

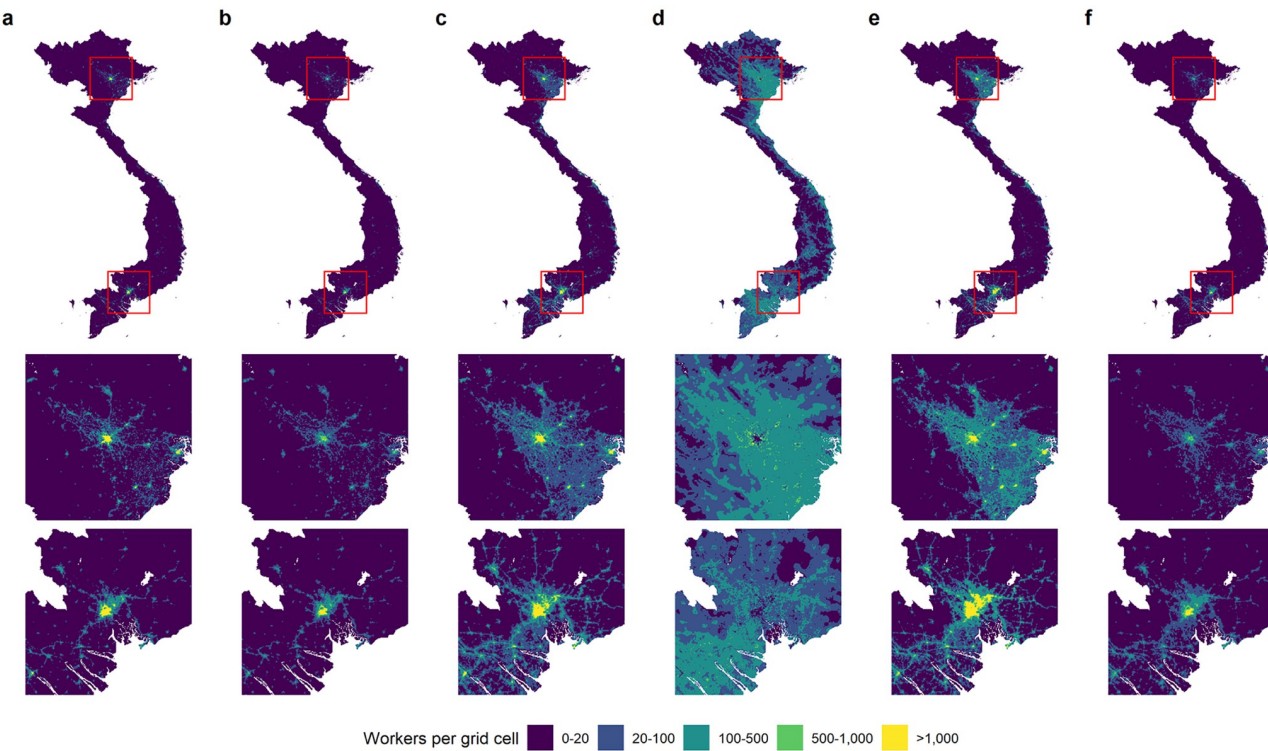

**Fig 2.** Spatial predictions for the number of (a) Managers and professionals, (b) Technicians and associate professionals, (c) Clerks and service workers, (d) Agricultural workers, (e) Craft workers and operators and (f) Elementary occupations per grid cell. Bottom panels depict the spatial distribution of the number of workers around Hanoi and Ho Chi Minh City, respectively. An interactive version can be accessed at https://shiny.wur.nl/occupation-map-vnm.

large in rural areas (50–100%) (S4 Fig). The spatial distribution of the other five occupation groups is mainly concentrated in urban and semi-urban areas. Large numbers of clerks and service workers, and craft workers and operators are depicted in the centers of the large cities but also in the industrial and populated areas that are spread around Hanoi and Ho Chi Minh City, as well as the urban coastal regions. This finding is to be expected as these two occupation categories cover the majority of jobs in the manufacturing and service sectors that employ most workers in industrial and populated areas in Vietnam. In contrast, the distribution of managers and professionals, and technicians and associate professionals is much more concentrated, with a very large presence in the centers of Hanoi and Ho Chi Minh City. These findings can be explained by the fact that high-skilled jobs are mainly concentrated in the head offices of large companies and ministries that are located in the center of major cities. In relative terms, the two high-skilled occupation categories are observed in all (semi)urban areas but only make up very small shares of the total labor force (S1 and S4 Figs). Elementary occupations contain a diverse group of jobs, including both unskilled agricultural and non-agricultural workers and therefore might be allocated in rural and urban areas. The maps show that, at least in Vietnam, elementary occupations mostly seem to be of the urban type (e.g. cleaners, hand packers and street vendors).

## Model evaluation

Fig 3 summarizes the performance of the seven super learner models that were run for each variable. The amount of variation ($R^2$) explained by the super learner model ranged from 0.63 for elementary occupations to 0.94 for agricultural workers. A comparison between the super learners and selected member models showed that the ensemble approach outperformed (n = 6), or was comparable (n = 1) to the predictions of the individual models with the highest RMSE. In all seven super learner models that were fitted, xgboost was the best performing machine learning algorithm, followed by random forest (n = 4), neural network (n = 2) and polynomial support vector machines (n = 1) (Table S3-S9 in S1 File). Predictions were located around the 1:1 line and the mean error was close to zero, indicating that the models are not biased.

We also investigated if the models were able to adequately reproduce the number of workers at higher subnational aggregations. To do this, we aggregated the predictions for the number of workers at grid cell level and compared them with the district-level number of workers that can be derived from the model input data for each occupation category (S2 Fig). A strong relationship between the two values indicated that the models are producing realistic values on average. Grid cell predictions were based on all predictor values, not only the district-level median values that were used to train the model, and therefore are likely to include observations at the tails of the distribution (e.g. remote areas). If the models would structurally under- or overperform for such observations, this would result in poor aggregated predictions and implies the models might be biased. The high $R^2$ of 0.85–0.95 (S6 Fig) between model outcomes and observed subnational statistics suggested that this is not the case.

To provide an indication of where the super learners were accurate and where they were not, we calculated the prediction error at grid cell level (S5 Fig). The maps showed that the errors were the most prominent in the regions where the occupation shares were very small, or even zero (i.e. agricultural workers in urban areas and non-agricultural workers in rural areas) and where the models were used to extrapolate on the edges of the training space.

To investigate the importance of the predictor variables on the model outcomes, we conducted a variable importance analysis to investigate which predictors contributed the most to

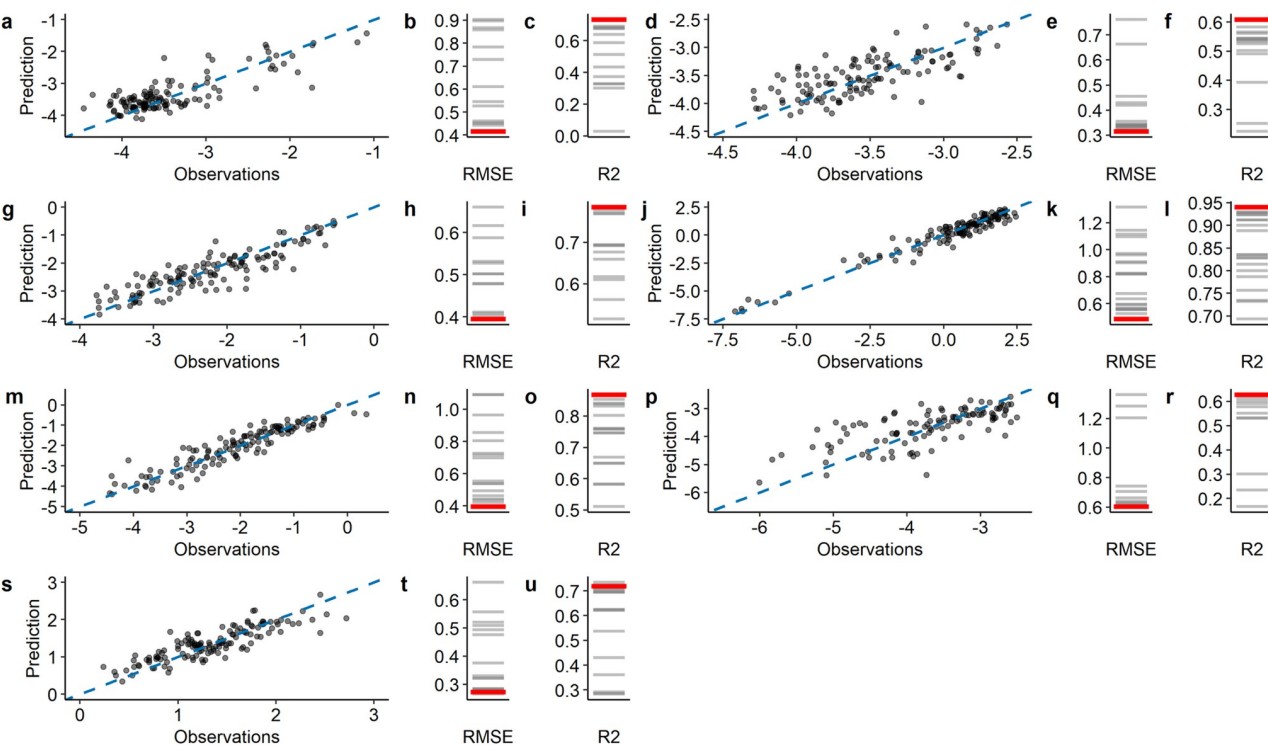

**Fig 3.** Performance of the super learners for the seven target variables: (a-c) Managers and professionals, (d-f) Technicians and associate professionals, (g-i) Clerks and service workers, (j-l) Agricultural workers, (m-o) Craft workers and operators, (p-r) Elementary occupations and (s-u) Labor force participation rate. The large panels compare the district-level observations with the super learner predictions for the hold-out dataset (n = 136). The dashed blue line represents the 1:1 line. Small panels show the RMSE and the $R^2$ for the super learners (red lines) and member models (gray lines) for the hold-out dataset. Predictions and observations are logit transformed values.

the model outcomes (Fig S7-S13 in S2 File) and we generated accumulated local effects plots to get insight on how they were related (Fig S14-S20 in S3 File). Urbanization was a key predictor of occupation share as the related predictor (urbpx_prp_5) was consistently ranked among the most important variables that explained the model results. Several predictors related to land use and economic activity frequently featured as explanatory variables in some of the occupation models. Not surprisingly, for agricultural workers, the distance to cropland (esac-cilc_dst011) and climate (bio_6) were also important determinants. In the models for managers and professionals distance to ports and airports were frequently included, which might indicate that high-skilled workers tend to be located in places that are internationally connected. Predictors related to the location of industrial activity (iron_steel, power, mining and cement) showed up as important for several member models but, perhaps in contradiction to expectations, and with the exception of mining, were not uniquely related to the location of craft workers and operators and technicians and associate professionals, of which a large share were expected to live close to these facilities.

## Relationship with spatial wealth

To evaluate the accuracy of our model estimates at grid level, we also compared our results with an exogenous dataset that presents comparable or related information [13, see 40 for a comparable approach to evaluate crop distribution maps]. Various studies showed that wealth is strongly correlated with income [e.g. 41], which, in turn, is determined for a large part by

labor income, and, hence, type of occupation. We therefore also expected to find a correlation between our gridded occupations maps and a global wealth index [13], which shows the spatial distribution of wealth at a resolution of 2.4 x 2.4 km. The wealth index is based on a machine learning model that used satellite imagery, mobile phone networks and connectivity data from Facebook as predictors and asset information from the Demographic and Health Surveys (DHS) as target variables, covering the period 2010–2018. This period does not overlap with our base year of 2009, but as the reallocation of labor across sectors is a long-run and gradual process [42], we expected that our occupation maps will also be representative for the period covered by the wealth map. The information in the DHS was collected by the U.S. Agency for International Development and is therefore independent of the population census that we used to train the super learners.

The relationship between wealth and occupation was confirmed by Fig 4, which shows that the proportion of low-skilled occupations, composed of agricultural workers ($R^2 = 0.39$), craft and operators ($R^2 = 0.43$), clerks and service workers ($R^2 = 0.29$) and elementary occupations ($R^2 = 0.28$), which make up 91% of the labor force (S1 Fig), explained a moderate part of relative wealth at grid-cell level. Wealth tends to be higher in (semi-)urban areas and therefore was negatively correlated with higher shares of agricultural workers, while a positive correlation with wealth was found for the other low-skilled occupations. There was a limited correlation between the wealth index and the share of managers and professionals, and technicians and associate professionals. A possible reason for this might be the limited coverage of (wealthy) high-skilled workers in the DHS. Another explanation could be the uneven distribution of wealth between high- and low-skilled workers within (urban) grid cells, which is not picked up by an average measure such as the wealth index.

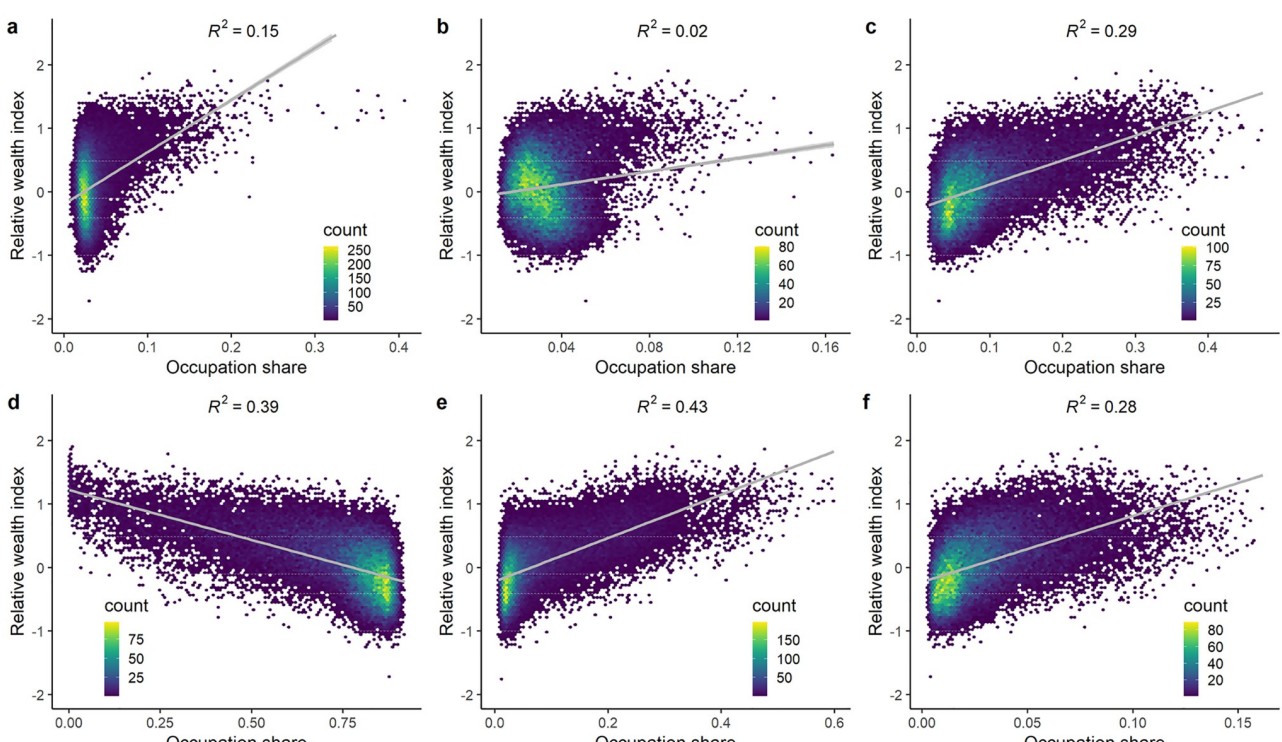

**Fig 4.** Scatterplot of the relative wealth index from [13] against the predicted occupation shares for (a) Managers and professionals, (b) Technicians and associate professionals, (c) Clerks and service workers, (d) Agricultural workers (e) Craft workers and operators and (f) Elementary occupations.

## Discussion

This paper presented an approach to downscale subnational labor statistics into high-resolution gridded maps using a super learner machine learning algorithm. The results showed that the predictions of the super learners were more accurate or at least comparable with those of the best-performing member models. Potential disadvantages of using ensemble approaches are the higher computational costs and complexity vis-à-vis fitting only a single type of machine learning model. Nonetheless, with the growing availability of high computing power, we expect this approach to be increasingly adopted by researchers with interest in the spatial prediction of social and biophysical indicators [e.g. 43].

This research can be extended, refined and improved in several directions. First, the predictive power of the super learners might be increased by collecting and incorporating additional spatial predictors. For example, the model for managers and professionals, which had relatively poor validation metrics, might benefit from spatial layers that show the distance or travel time to schools, hospitals, city halls and other government buildings, which are typical work locations for this type of occupation [see 14 for an example involving the distance to health facilities]. Similarly, the accuracy of the craft workers and operators model might be improved by adding data on the location of industrial zones and factories. Unfortunately, digital maps with this type of information are not available for Vietnam. Overall, the predictive power of spatial indicators that measure the distance to work will probably be lower for high-skilled workers as they tend to commute over longer distances in comparison to low-skilled workers [24, 25].

Second, in the future we aim to apply our approach to larger regions, such as South-East Asia and the world, resulting in a product that complements existing global maps with socio-economic indicators, such as population (www.worldpop.org), education [10] and wealth [13]. For a substantial number of countries, the required subnational labor statistics can be extracted from the population census, which is available by means of the IPUMS database [30]. Another source of information is summary reports of national labor force surveys that are regularly published by national statistical agencies. The level of detail, however, can differ considerably between data sources and countries. In the case of Vietnam, we were able to obtain information for second-level administrative units. For many other countries, only a few data points, representing first-level administrative units or broad regions, might be available. Combining the data of a large number of countries will make it possible to train machine learning models that achieve higher accuracy and can be used to generate plausible out-of-sample predictions for countries for which no data is available [13, 15].

Finally, an interesting avenue of future research would be to investigate the possibility of implementing our approach to derive high-resolution maps for other (labor) indicators, such as gender-specific occupation maps. As mentioned above, differences in labor force participation between men and women have a strong spatial dimension, which can be disentangled by our analytical approach. Such an indicator would also inform spatial predictions of the proportion of women in managerial positions, which is one of the SDG5: Gender equality indicators. Another relevant indicator is the unemployment rate, listed under SDG8: Decent work and economic growth. However, mapping unemployment rates is complex because it is the sum of frictional, cyclical, and structural components [44]. In advance, it is not clear which of these components will be captured by the various spatial predictors. For example, night lights are a proxy for economic activity [45] and therefore might pick up the sum of all three components, while the distance to cropland, airports and industrial facilities might be correlated with structural unemployment rates. Although our analysis focused on labor data, it can equally be applied to other statistics that are only available at the subnational level, including sectoral economic output, health and education information [46, 47].

## Supporting information

**S1 Fig. Total labor force and occupation shares in Vietnam for the year 2009.** Source: Minnesota Population Center. Integrated Public Use Microdata Series, International: Version 7.2 [dataset]. Minneapolis, MN: IPUMS; 2019. https://doi.org/10.18128/D020.V7.2 for labor force participation rate and occupation shares, and Pezzulo et al. (2017) for working age population.
(PNG)

**S2 Fig.** District-level information: occupation shares for (a) Managers and professionals, (b) Technicians and associate professionals, (c) Clerks and service workers, (d) Agricultural workers, (e) Craft workers and operators, (f) Elementary occupations, (g) Labor force participation rate and (h) Working age population in Vietnam for the year 2009. Source: Minnesota Population Center. Integrated Public Use Microdata Series, International: Version 7.2 [dataset]. Minneapolis, MN: IPUMS; 2019. https://doi.org/10.18128/D020.V7.2 for labor force participation rate and occupation shares, and Pezzulo et al. (2017) for working age population.
(PNG)

**S3 Fig. Correlation matrix for predictors before normalization and Yeo-Johnson power transformation.** After normalization and Yeo-Johnson power transformation, we used step_corr(., threshold = .7) from the R recipes package to remove all predictors with an absolute correlation equal or larger than 0.7. Consequently, 14 predictors were removed from the analysis (bio_1, bio_5, bio_6, dmsp, dst_bsgmi, dst_ghslesaccilcguf, esaccilc_dst040, int_airports, osm_dst_road, osm_dst_roadintersec, srtm_slope, srtm_topo, travel_time, viirs), leaving 18 predictors that were used as final input.
(PNG)

**S4 Fig.** Super learner results for (a) Managers and professionals, (b) Technicians and associate professionals, (c) Clerks and service workers, (d) Agricultural workers, (e) Craft workers and operators, (f) Elementary occupations and (g) Labor force participation rate.
(PNG)

**S5 Fig.** Prediction errors for (a) Managers and professionals, (b) Technicians and associate professionals, (c) Clerks and service workers, (d) Agricultural workers, (e) Craft workers and operators, (f) Elementary occupations and (g) Labor force participation rate. Prediction errors are logit transformed values, which are provided with a probability of 67%, which is the 1 standard deviation upper and lower prediction interval.
(PNG)

**S6 Fig.** District-level comparison between observations and super learner predictions for (a) Managers and professionals, (b) Technicians and associate professionals, (c) Clerks and service workers, (d) Agricultural workers, (e) Craft workers and operators and (f) Elementary occupations. Dashed blue line represents the 1:1 line. Solid blue line indicates the regression line, with 95% confidence intervals in grey. District-level observations on the number of workers are calculated by multiplying district-level data on occupation share, labor force participation and working age population (aggregated from grid-level values), depicted in S2 Fig.
(PNG)

**S1 Table. Main occupation categories based on the International Classification of Occupations 08 (ILO 2012).**
(PDF)

**S2 Table. List of selected predictors for the machine learning models.**
(PDF)

**S1 File. Hyperparameters for super learner model members, sorted by RMSE.**
(PDF)

**S2 File. Variable importance plots.** Results are presented for the five super learner model members with the highest weight and for the top 10 predictors. Starting values for the horizontal bars indicate the RMSE for the full model. Predictors with the largest bars are the most important because permuting them results in higher RMSE. Error bars indicate results for 10 different permutations.
(PDF)

**S3 File. Accumulated local effects.** Results are presented for the five super learner model members with the highest weight.
(PDF)

## Acknowledgments

We would like to thank Tom Hengl for advice on how to calculate the prediction errors.

## Author Contributions

**Conceptualization:** Michiel van Dijk.

**Data curation:** Michiel van Dijk, Thijs de Lange.

**Formal analysis:** Michiel van Dijk, Thijs de Lange.

**Funding acquisition:** Michiel van Dijk.

**Methodology:** Michiel van Dijk, Thijs de Lange, Paul van Leeuwen, Philippe Debie.

**Project administration:** Michiel van Dijk.

**Software:** Michiel van Dijk, Thijs de Lange.

**Visualization:** Michiel van Dijk, Thijs de Lange.

**Writing – original draft:** Michiel van Dijk, Thijs de Lange.

**Writing – review & editing:** Michiel van Dijk, Thijs de Lange, Paul van Leeuwen, Philippe Debie.

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
