## [Decision Letter · Decision Letter 0]

26 May 2022

PONE-D-22-10590Occupations on the map: Using a super learner algorithm to downscale labor statisticsPLOS ONE

Dear Dr. van Dijk,

Thank you for submitting your manuscript to PLOS ONE. After careful consideration, we feel that it has merit but does not fully meet PLOS ONE’s publication criteria as it currently stands. Therefore, we invite you to submit a revised version of the manuscript that addresses the points raised during the review process.

We look forward to receiving your revised manuscript.

Kind regards,

Sotirios Koukoulas, Ph.D

Academic Editor

PLOS ONE

Journal Requirements:

2. We note that Figures 1, 2, S2-S11 in your submission contain map/satellite images which may be copyrighted. All PLOS content is published under the Creative Commons Attribution License (CC BY 4.0), which means that the manuscript, images, and Supporting Information files will be freely available online, and any third party is permitted to access, download, copy, distribute, and use these materials in any way, even commercially, with proper attribution. For these reasons, we cannot publish previously copyrighted maps or satellite images created using proprietary data, such as Google software (Google Maps, Street View, and Earth). For more information, see our copyright guidelines: http://journals.plos.org/plosone/s/licenses-and-copyright.

a) You may seek permission from the original copyright holder of Figures 1, 2, S2-S11 to publish the content specifically under the CC BY 4.0 license.  

Natural Earth (public domain): http://www.naturalearthdata.com/.

Reviewers' comments:

Reviewer's Responses to Questions

**Comments to the Author**

1. Is the manuscript technically sound, and do the data support the conclusions?

Reviewer #1: Partly

Reviewer #2: Yes

2. Has the statistical analysis been performed appropriately and rigorously? 

Reviewer #1: Yes

Reviewer #2: Yes

3. Have the authors made all data underlying the findings in their manuscript fully available?

Reviewer #1: Yes

Reviewer #2: Yes

4. Is the manuscript presented in an intelligible fashion and written in standard English?

Reviewer #1: Yes

Reviewer #2: Yes

5. Review Comments to the Author

Reviewer #1: My opinion is that the paper could be accepted with major revisions by authors considering the following comments.

Major comments. The methodology is not clearly presented and as such, is not convincing. Equally important, how the output gridded map is compared with the actual occupational status is vague and probably incorrect. For this reason, it has to be refined.

Here are my comments in detail:

1. Introduction. The paper lacks a paragraph explaining which is the originality of this paper in terms of the methodological approach. Is it just the use of a super learner, or something more, like designing the gridded map? This should be stated more clearly.

2. Line 31-33. I suggest you include more recent papers regarding the spatial uneven distribution of labor and expand a little more on the subject. For example, include the following recent works. Strumsky, et al., 2021. “As different as night and day: Scaling analysis of Swedish urban areas and regional labor markets." Environment and Planning B: Urban Analytics and City Science. Grekousis et al., 2018 “More Flexible Yet Less Developed? Spatio‑Temporal Analysis of Labor Flexibilization and Gross Domestic Product in Crisis‑Hit European Union Regions”. Social Indicators Research.

3. Page 6, Line 89. The labor force includes the unemployed. How is equal to the sum of all workers? Rewrite and add citation for the labor force definition.

4. Line 105 … of each grid cell. Of which grid? Authors should mention clearly how they selected the grid (e.g., based on other grid data readily available, or just designed by themselves?) How many cells has this grid?

5. Line 138. Even 0.7 is highly likely to bring multicollinearity issues. I think authors should remove variables correlated at 0.7 level or higher. For example, I would expect viirs with dmsp to be highly correlated. What’s their correlation?

6. Line 138. Which variables were removed, and how many did you finally keep? Table S2 just mentions the 32 predictors. It should be clear which variables were used for training.

7. Line 162. From my understanding authors trained a set of 538 objects, tested over 136 objects on an unknown (not specified in their text) number of variables to make a prediction for some thousands (I believe) cells. Authors should clearly explain their methodology process. For example, which are the specific variables used as predictors for each different target variable? However, I think that if they used the above approach, the results may not be accurate. The reason is that occupational data were downscaled from only 674 districts to some thousands (I guess) and then used again for prediction at this lower scale. Actually, neural networks are kind of prone to errors when the dataset is not large.

8. Line 210. TablesS3-S9. Authors should use a standard way of presenting results in TablesS3-S9 making thus easier for other researchers to delve deeper into the details of the adopted architecture and the performance of the ANNs and the other machine learning methods used. Authors are strongly recommended to use and cite a reporting scheme proposed by Grekousis 2019 “Artificial neural networks and deep learning in urban geography: A systematic review and meta-analysis.” Computers Environment and Urban Systems. Authors should use Table 4 of the above paper at least for the neural networks they already report and adapt it according to their needs (for example, as this is an ensemble approach, they don’t have to provide any graphs).

9. Line 212. This is like a loop. Authors begin from generalized data (small dataset), downscale them, and then generalize again. It is expected that prediction error is likely to decrease when you get back to the near original scale. This approach is not convincing unless they can provide more evidence. Authors should better explain or follow another path. For example, it would be more accurate if authors could find occupational data for a higher (more detailed) administrative level than the second level they used, for let’s say 100 randomly selected cells (each cell could enclose more than one of the more detailed administrative level). As the occupational data come from the national census, it’s expected that data should exist. Then, authors could just compare the predicted values at each cell, with the actual values at the overlapping administrative units.

10. Line 218. How exactly is this error calculated? What is the unit of the error mapped at fig S11 (e.g., std dev?)

11. Line 226. Esaccilc_dist190 does not exist in Table S2

12. Section3.3. I think this section is not well presented and not convincing. Authors try to use a different dataset to compare their output. First, the wealth index spans in a long period (2010-2018) outside the reference study time (2009). Second, the R2 is low (below 0.50) in all cases, something also clearly seen in Figure 4, so a conclusion of a good fit is an overstatement. I would suggest dropping this section as it creates more confusion than convincing evidence for the model’s accuracy.

Reviewer #2: Dear Editor,

I have carefully read the manuscript with title "Occupations on the map: Using a super learner algorithm to downscale labor statistics" which concerns the implementation of a generic machine learning approach (the super learner algorithm) in constructing a map illustrating occupation in the desired (depending also on the availabel data) spatial detail. The machine learning tools employed for the statistical analysis are well enstablished (relying mostly on already developed packages in R) and the results are both interesting and interpretable. The novelty of this paper is considerable from the econometrics/geographical perspective and it mainly concerns the application of a flexible ML procedure in analysing sampled data in various spatial scales in order to construct/recover the occupation map in a region. The manuscript is well written and the interested reader can easily follow it. Therefore, I recommend the current paper for publication.

Some (very) minor concerns/suggestions:

- p.4, l.85: i should be a subscript

- p.7, l.138: "we removed all variables that had a correlation of 0.9 or larger" You could try also a Ridge-type estimation scheme to keep all the available information. You may at least mention this approach as a possible alternative in order to keep the same set of predictors for all of you modelling operations to avoid inconsistencies in the interpretation.

- p.7, l.139: It would be nice to provide a small description on how the Super Learner procedure works. It would be excellent if you could provide a graphical representation of this procedure to your problem illustrating the various ML methods that you are using (Random Forests, Logistic model, etc).

- You might also provide some rough estimates for the training costs in CPU time if it is possible.

6. PLOS authors have the option to publish the peer review history of their article (what does this mean?). If published, this will include your full peer review and any attached files.

Reviewer #1: No

Reviewer #2: No

---

## [Author Response · Author response to Decision Letter 0]

27 Oct 2022

Dear Reviewers,

Thank you very much for the useful comments. We believe these comments are relevant and therefore we have made an effort to respond to them in the best possible way. As a consequence the paper has improved a lot. We sincerely hope that our changes are in line with the expectations of the reviewers and the paper can be accepted for publication.

In the remainder of this letter, we provide a point-by-point response to the various comments. We do this separately for the comments made by each reviewer. Original comments are in black and our response is in blue. Where relevant, line numbers are added to better trace the changes made to the document. We also made some other changes following the comments of the PLOS ONE editor related to copyright of figures, which are placed at the end of the document. 

Best wishes,

Michiel van Dijk (on behalf of all authors). 

Reviewer #1

Major comments. The methodology is not clearly presented and as such, is not convincing. Equally important, how the output gridded map is compared with the actual occupational status is vague and probably incorrect. For this reason, it has to be refined.

Here are my comments in detail:

1. Introduction. The paper lacks a paragraph explaining which is the originality of this paper in terms of the methodological approach. Is it just the use of a super learner, or something more, like designing the gridded map? This should be stated more clearly.

Thanks for the comment. Our paper makes two contributions to the literature: (1) We present an approach to create gridded maps for labor statistics. Labor is one if the key factors of production and therefore detailed spatially explicit statistics are key information for (local) policy makers. Labor statistics are only available at the national level and sporadically at the subnational level. We provide an approach to create fine scale gridded maps of labor statistics to better inform policies (2) the existing literature on downscaling of subnational indicators as well as the related literature on spatial extrapolation of geo-coded information (e.g. for soils) has mainly relied on the use of a single ML approach (mostly random forest), which might lead to biased or sub-optimal results. Our paper uses an ensemble approach (i.e. a super learner) to address this issue. 

To emphasize this better, we rearranged the introduction and clearly stated our contribution in the beginning of the introduction L48-56: “The contribution of this paper is to demonstrate an approach to downscale subnational labor statistics to a fine-scale spatial grid using machine learning approaches. As such this paper contributes to a rapidly expanding literature, which uses machine learning and advanced statistical models to create fine-scale gridded maps of socio-economic indicators. Key examples include the mapping of population (Leyk et al. 2019), educational attainment (Graetz et al. 2019), child growth (Osgood-Zimmerman et al. 2018), poverty (Pokhriyal and Jacques 2017) and wealth (Chi et al. 2022). To the best of our knowledge, this is the first application to apply these techniques to downscale labor data.”

 2. Line 31-33. I suggest you include more recent papers regarding the spatial uneven distribution of labor and expand a little more on the subject. For example, include the following recent works. Strumsky, et al., 2021. “As different as night and day: Scaling analysis of Swedish urban areas and regional labor markets." Environment and Planning B: Urban Analytics and City Science. Grekousis et al., 2018 “More Flexible Yet Less Developed? Spatio Temporal Analysis of Labor Flexibilization and Gross Domestic Product in Crisis Hit European Union Regions”. Social Indicators Research.

Thanks for the suggestions. We have added the references in L38-39. 

3. Page 6, Line 89. The labor force includes the unemployed. How is equal to the sum of all workers? Rewrite and add citation for the labor force definition.

The labor force definition as well as all other definitions were taken from the ILO and we added the reference on L122-123. The confusion is caused by the term ‘workers’, which is not clearly defined and might suggest this only included the number of employed people. Census and labour force surveys ask questions on type of occupation to all people that are in the labour force, either employed or unemployed. We removed the confusing sentence and added a clarification in L142-143. The census included questions for each person in the labor force on his/her type of occupation using the the detailed ISCO 08 classification.

4. Line 105 … of each grid cell. Of which grid? Authors should mention clearly how they selected the grid (e.g., based on other grid data readily available, or just designed by themselves?) How many cells has this grid?

Indeed, this is not clear. In this case we simply used an existing product of Pezzulo et al (2017), which presents a gridded map with population data broken down by age classes. We also explained how this map was calculated by the authors, which might have created confusion as it seemed that we made the calculations. We added the following lines (L148-152) to make this clear: “To calculate the working age population at each grid cell, we used population maps with 5-year age group compositions from Pezzulo et al. (2017). The main data source for this product was the 2009 Vietnam population census, which we also used as the main input.”

5. Line 138. Even 0.7 is highly likely to bring multicollinearity issues. I think authors should remove variables correlated at 0.7 level or higher. For example, I would expect viirs with dmsp to be highly correlated. What’s their correlation?

We implemented the proposal of the reviewer and reran our analysis with a maximum correlation of 0.7. As a result, 14 out of the 32 predictors were removed from the analysis (before only 4 but this was not well described). We added a correlation matrix in the SI and describe clearly which variables were kept for the analysis. We also mention that 14 variables are removed in the main text L185-186. As a consequence, the outcomes have changed somewhat but are overall very similar to the ones presented in our first submission.

6. Line 138. Which variables were removed, and how many did you finally keep? Table S2 just mentions the 32 predictors. It should be clear which variables were used for training.

See response to previous comment.

7. Line 162. From my understanding authors trained a set of 538 objects, tested over 136 objects on an unknown (not specified in their text) number of variables to make a prediction for some thousands (I believe) cells. Authors should clearly explain their methodology process. For example, which are the specific variables used as predictors for each different target variable? However, I think that if they used the above approach, the results may not be accurate. The reason is that occupational data were downscaled from only 674 districts to some thousands (I guess) and then used again for prediction at this lower scale. Actually, neural networks are kind of prone to errors when the dataset is not large.

This comment is not clear to us. The reviewer thinks that we ‘somehow’ downscaled the subnational data to the grid cell and used this for prediction. This is not the case. We ran all machine learning models, including the super learner, which simply combines the individual models, at the level of the subnational units (538 objects for training and 136 objects for testing) by combining data of the target variable at the subnational value with the median of grid cell data that is located in the subnational areas (177-179). As explained on L57-58, this approach is identical Stevens et al. (2015) and Nicolas et al. (2016), who created gridded population and livestock maps, respectively. To make this clearer we added the following sentence in L201-203: “All six machine learning algorithms and the super learner models used data at the subnational level for training and testing (i.e. 674 data points in total).”

8. Line 210. TablesS3-S9. Authors should use a standard way of presenting results in TablesS3-S9 making thus easier for other researchers to delve deeper into the details of the adopted architecture and the performance of the ANNs and the other machine learning methods used. Authors are strongly recommended to use and cite a reporting scheme proposed by Grekousis 2019 “Artificial neural networks and deep learning in urban geography: A systematic review and meta-analysis.” Computers Environment and Urban Systems. Authors should use Table 4 of the above paper at least for the neural networks they already report and adapt it according to their needs (for example, as this is an ensemble approach, they don’t have to provide any graphs).

We agree it is important to add detail on the machine learning approaches used, assumptions and hyperparameters selected. For this reason, we added Tables S3-S9, which provide the main hyperparameters of the most important models in our super learner. With respect to the neural network, it provides additional detail on the three parameters that can be tuned with the R neural network package we used for the analysis (nnet, Venables and Ripley, 2002). The elaborate reporting scheme proposed by Grekousis also lists several other parameters, which are, however, not relevant (i.e. tunable) for the nnet package, e.g. regularization coefficient and dropout. Moreover, as we ran a very large number of models in an ensemble framework, it is not practically feasible to add all the details in the form of tables in an annex. Instead, we prefer to make our analysis fully reproducible by adding code and data in open repositories (see code and data statement in the papers). In this way the interested reader can extract all information possible, even going beyond the reporting scheme of Grekousis (2019). In addition, we added an extra paragraph in the SI, that lists the R packages used for each of the machine learning models, which makes it easy for interested readers to look for additional information on hyperparameters etc. and stresses the need to provide clarity about packages used and hyperparameters selected in the spirit of Grekousis (2019).

9. Line 212. This is like a loop. Authors begin from generalized data (small dataset), downscale them, and then generalize again. It is expected that prediction error is likely to decrease when you get back to the near original scale. This approach is not convincing unless they can provide more evidence. Authors should better explain or follow another path. For example, it would be more accurate if authors could find occupational data for a higher (more detailed) administrative level than the second level they used, for let’s say 100 randomly selected cells (each cell could enclose more than one of the more detailed administrative level). As the occupational data come from the national census, it’s expected that data should exist. Then, authors could just compare the predicted values at each cell, with the actual values at the overlapping administrative units.

We agree with the reviewer that using more detailed subnational information would be best for validation. Unfortunately, the micro data from the population census provided by IPUMS is only representative at 2nd level so we cannot calculate labor force statistics at finer subnational levels. Despite of this, we do think that our current approach is useful for validation because it says something about the bias of the model(s). Grid cell predictions are based on all predictor values (e.g. the full distribution of observations) within a subnational unit, not only the median values that were used to train the model. As such it will also include values around the median, even including observations at the tails of the distribution (e.g. remote areas with near zero nightlight etc), which were most likely not observed in the training and testing datasets. If the model would structurally under- or overperform for such values, this would result in a strong bias. On the other hand, if we find a strong relationship between aggregated predictions and subnational census values, this is evidence that the model is producing realistic values on average (i.e. has a low bias). 

We added several sentences to make this point in 283-294. “We also investigated if the models were able to adequately reproduce the number of workers at higher subnational aggregations. To do this, we aggregated the predictions for the number of workers at grid cell level and compared this with the district-level number of workers that can be derived from the model input data for each occupation category (Figure S2). A strong relationship between the two values indicates that the models are producing realistic values on average. Grid cell predictions are based on all predictor values, not only the median values that were used to train the model, and therefore are likely to include observations at the tails of the distribution (e.g. remote areas). If the models would structurally under or over perform for such values, this would result in poor aggregated predictions and implies the models might be biased. The high R2 of 0.85-0.95 (Figure S6) between model outcomes and observed subnational statistics suggest this is not the case.”

10. Line 218. How exactly is this error calculated? What is the unit of the error mapped at fig S11 (e.g., std dev?)

The calculation of the error is explained in L227-231 but we now also added the unit (and in the SI as well): “We followed the approach used by Hengl et al. (2021) to calculate 67% (1 standard deviation) grid-cell prediction errors for the super learner models.”

11. Line 226. Esaccilc_dist190 does not exist in Table S2

This was a typo. As a consequence of the updated correlation coefficient restriction (see above), the list of included predictors has changed so we have updated this section accordingly. 

12. Section3.3. I think this section is not well presented and not convincing. Authors try to use a different dataset to compare their output. First, the wealth index spans in a long period (2010-2018) outside the reference study time (2009). Second, the R2 is low (below 0.50) in all cases, something also clearly seen in Figure 4, so a conclusion of a good fit is an overstatement. I would suggest dropping this section as it creates more confusion than convincing evidence for the model’s accuracy.

We do not agree with the remark of the reviewer. As explained in the text, there is a clear theoretical link between share of occupation in a region/grid cell and average wealth. One would expect to pick this up by means of a simple regression analysis. There are no standard definitions for a ‘low’ and ‘high’ R2 but in our opinion an R2 of between 0.5 and 0.2 is often considered as moderate and possibly even high in social sciences (note that an R2 of 0.5 is equal to a correlation of ~0.7!), but certainly not low. In fact, we think the finding of a simple R2 of around 0.4 for the two main occupation classes (agric and craft workers) which make up 73% of the labor force, is remarkable and confirms the validity of our approach. Also note that we run the regression on a large number of grid cells (~ 33.000) and therefore lower R2 are too be expected because of random errors. We suspect that the correlation would even be higher if we could add various control variables such as local taxes, education level etc but such an analysis would fall out of the scope of this paper, nor are the data available to do this. Finally, it is very common in the literature on downscaling to use external datasets for validation. Other examples are Chi et al. (2022) and Yu et al. (2020). We follow this tradition. Finally, it is correct that the wealth map and our product cover different periods. Change in occupational structure, however, is a long-run process related to structural change in the economy. It is therefore plausible to assume that the occupation map for 2009 is also representative for the period 2010-2018, which is covered by the poverty map, and therefore both can be compared. We modified this paragraph to explain these pojnts (341-344). “This period does not overlap with our base year of 2009, but as the reallocation of labor across sectors is a long-run and gradual process (Timmer, Vries, and Vries 2015), we expect that our occupation maps will also be representative for the period covered by the wealth map.”

 

Reviewer #2

Some (very) minor concerns/suggestions:

- p.4, l.85: i should be a subscript

Thanks for spotting this – Changed.

- p.7, l.138: "we removed all variables that had a correlation of 0.9 or larger" You could try also a Ridge-type estimation scheme to keep all the available information. You may at least mention this approach as a possible alternative in order to keep the same set of predictors for all of you modelling operations to avoid inconsistencies in the interpretation.

Reviewer #1 also had a remark about this assumption. We decided to follow his advice and use a correlation of 0.7 instead (see above).

- p.7, l.139: It would be nice to provide a small description on how the Super Learner procedure works. It would be excellent if you could provide a graphical representation of this procedure to your problem illustrating the various ML methods that you are using (Random Forests, Logistic model, etc).

We added the following paragraph (L188-193) to explain the concept of a super learner algorithm. “We used an ensemble approach, referred to as a super learner, to predict labor statistics at the grid level. A super learner is an algorithm that combines the results of multiple machine learning models or the same model with different parameters and settings. Predictions are then generated by weighting the outcomes of the individual member models. It has been demonstrated that predictions of a super learner have the same or higher accuracy in comparison to those generated by means of single machine learning models (Laan, Polley, and Hubbard 2007).” 

We have thought about adding some sort of graphical representation as suggested by the reviewer but are not fully convinced about the need to add this (nor do we have clear ideas on what to add). 

- You might also provide some rough estimates for the training costs in CPU time if it is possible.

This is an interesting idea but as we have used different machines to run our model. We are currently in the process of running the scripts on a high-compute cluster and therefore did not collect consistent data on CPU time and, hence, are unable to add this.

# PLOS ONE editor

We had a look at the templates and made changes accordingly.

2. We note that Figures 1, 2, S2-S11 in your submission contain map/satellite images which may be copyrighted. All PLOS content is published under the Creative Commons Attribution License (CC BY 4.0), which means that the manuscript, images, and Supporting Information files will be freely available online, and any third party is permitted to access, download, copy, distribute, and use these materials in any way, even commercially, with proper attribution. For these reasons, we cannot publish previously copyrighted maps or satellite images created using proprietary data, such as Google software (Google Maps, Street View, and Earth). For more information, see our copyright guidelines: http://journals.plos.org/plosone/s/licenses-and-copyright.

Figure 1 and S2 depict data from IPUMS. This data can be used for analysis and visualization as long as an appropriate reference is added, which we have done in the text. Figure 2, S10 and S11 present the results of our analysis so this can be presented without any problems. S3-S9 present maps of our predictors. Although all datasets are open access, we are not 100% sure if we can share all maps in this format. For this reason, we have removed these maps from the SI. As readers can easily recreate this data

---

## [Decision Letter · Decision Letter 1]

10 Nov 2022

Occupations on the map: Using a super learner algorithm to downscale labor statistics

PONE-D-22-10590R1

Dear Dr. van Dijk,

We’re pleased to inform you that your manuscript has been judged scientifically suitable for publication and will be formally accepted for publication once it meets all outstanding technical requirements.

Kind regards,

Sotirios Koukoulas, Ph.D

Academic Editor

PLOS ONE

Additional Editor Comments (optional):

Reviewers' comments:

Reviewer's Responses to Questions

**Comments to the Author**

1. If the authors have adequately addressed your comments raised in a previous round of review and you feel that this manuscript is now acceptable for publication, you may indicate that here to bypass the “Comments to the Author” section, enter your conflict of interest statement in the “Confidential to Editor” section, and submit your "Accept" recommendation.

Reviewer #1: All comments have been addressed

2. Is the manuscript technically sound, and do the data support the conclusions?

Reviewer #1: Yes

3. Has the statistical analysis been performed appropriately and rigorously? 

Reviewer #1: Yes

4. Have the authors made all data underlying the findings in their manuscript fully available?

Reviewer #1: Yes

5. Is the manuscript presented in an intelligible fashion and written in standard English?

Reviewer #1: Yes

6. Review Comments to the Author

Reviewer #1: All comments were addressed. The statistical analysis has been performed appropriately and rigorously. The paper is accepted with no other comments.

7. PLOS authors have the option to publish the peer review history of their article (what does this mean?). If published, this will include your full peer review and any attached files.

Reviewer #1: No

---

## [Editor Report · Acceptance letter]

29 Nov 2022

PONE-D-22-10590R1 

Occupations on the map: Using a super learner algorithm to downscale labor statistics 

Dear Dr. van Dijk:

I'm pleased to inform you that your manuscript has been deemed suitable for publication in PLOS ONE. Congratulations! Your manuscript is now with our production department. 

Kind regards, 

on behalf of

Dr. Sotirios Koukoulas 

Academic Editor

PLOS ONE